# Conditional permeabilization of the *P. falciparum* plasma membrane in infected cells links cation influx to reduced membrane integrity

**Mariame Sylla**[¤a☯], **Ankit Gupta**[¤b☯], **Jinfeng Shao**[☯], **Sanjay A. Desai**[ID]*

Laboratory of Malaria and Vector Research, National Institute of Allergy and Infectious Diseases, National Institutes of Health, Rockville, MD, United States of America

☯ These authors contributed equally to this work.
¤a Current address: Brigham and Women's Hospital, Harvard Medical School, Boston, MA, United States of America
¤b Current address: All India Institute of Medical Sciences (AIIMS), Munshiganj, Raebareli, U.P., India
* sdesai@niaid.nih.gov

**Data Availability Statement:** All relevant data are within the paper and its Supporting Information files.

## Abstract

The intracellular human malaria parasite, *Plasmodium falciparum*, uses the PfATP4 cation pump to maintain $Na^+$ and $H^+$ homeostasis in parasite cytosol. PfATP4 is the target of advanced antimalarial leads, which produce many poorly understood metabolic disturbances within infected erythrocytes. Here, we expressed the mammalian ligand-gated TRPV1 ion channel at the parasite plasma membrane to study ion regulation and examine the effects of cation leak. TRPV1 expression was well-tolerated, consistent with negligible ion flux through the nonactivated channel. TRPV1 ligands produced rapid parasite death in the transfectant line at their activating concentrations, but were harmless to the wild-type parent. Activation triggered cholesterol redistribution at the parasite plasma membrane, reproducing effects of PfATP4 inhibitors and directly implicating cation dysregulation in this process. In contrast to predictions, TRPV1 activation in low $Na^+$ media accentuated parasite killing but a PfATP4 inhibitor had unchanged efficacy. Selection of a ligand-resistant mutant revealed a previously uncharacterized G683V mutation in TRPV1 that occludes the lower channel gate, implicating reduced permeability as a mechanism for parasite resistance to antimalarials targeting ion homeostasis. Our findings provide key insights into malaria parasite ion regulation and will guide mechanism-of-action studies for advanced antimalarial leads that act at the host-pathogen interface.

## Introduction

Malaria, caused by five species of *Plasmodium* parasites in humans, remains a public health priority, with *P. falciparum* and *P. vivax* the predominant causes of morbidity and mortality [1]. Because there is not an effective vaccine and the parasite can develop resistance to available

**Funding:** SAD: This work was supported by the Intramural Research Program of the National Institutes of Health, National Institute of Allergy and Infectious Diseases. The funders had no role in study design, data collection and analysis, decision to publish, or preparation of the manuscript.

**Competing interests:** The authors have declared that no competing interests exist.

drugs [2], there has been a concerted effort to develop new drugs. The not-for-profit Medicines for Malaria Venture (MMV) has engaged with both basic scientists and the pharmaceutical sector to develop new and effective antimalarials.

With this goal in mind, several high-throughput screens of chemical libraries have been carried out using whole-cell proliferation assays and cultured *P. falciparum* parasites [3, 4]. The large number of scaffolds identified were subsequently filtered to include action on multidrug-resistant parasite lines and to remove agents having mammalian cell cytotoxicity, yielding distinct scaffolds for the development of new antimalarial drugs. Remarkably, target identification studies using three of the most exciting scaffolds—spiroindolones, (+)-SJ733, and pyrazoleamides [5–7]—suggest that all of these agents work primarily against PfATP4, a P-type ATPase transporter localizing to the parasite plasma membrane (PPM) [5].

Because identification of the PfATP4 target used *in vitro* selection of resistant mutants followed by whole-genome sequencing, the ions or solutes transported by PfATP4 were unclear and have been pursued with independent biochemical studies. In contrast to the host erythrocyte membrane and the parasitophorous vacuolar membrane (PVM), the PPM cannot presently be studied with patch-clamp methods [8], necessitating indirect assays for PfATP4 activity. The first report used PfATP4 expression in *Xenopus* oocytes and measurement of $Ca^{++}$-dependent ATPase activity to propose that PfATP4 functions as a $Ca^{++}$ efflux pump [9]. More recently, however, detergent-freed parasites loaded with ion-sensitive indicator dyes have been used to support a $Na^+/H^+$ exchanger that pumps $Na^+$ out and $H^+$ into the parasite compartment [6, 7, 10, 11].

Each of the PfATP4 inhibitor scaffolds kills parasites rapidly and at low concentrations, producing changes in the intracellular parasite morphology [6]. These compounds also cause the PPM to become sensitive to saponin [12], a cholesterol-requiring detergent that permeabilizes the host membrane but not the PPM in healthy parasites with uninhibited PfATP4 activity. Whether these changes result directly from PfATP4 inhibition or from the downstream effects of disrupted ion homeostasis in the parasite remains unclear.

To bypass some of the limitations associated with flux measurements on detergent-freed parasites and to explore how changes in ion homeostasis alter the intracellular parasite's physiology, we have developed and implemented a novel strategy based on expression of a mammalian ion channel on the PPM. Because expression of most mammalian channels at the PPM would lead to dysregulated ion flux and be toxic to the parasite, transfectant lines carrying these channels are not expected to be viable. We avoided this concern by use of a ligand-gated channel, TRPV1, which permits ion flux upon only upon activation by specific ligands [13]. We found that capsaicin, a commonly used TRPV1 ligand, and a potent derivative are nontoxic to wild-type parasites but rapidly kill transfected parasites expressing TRPV1, establishing the need for ion regulation at this membrane without use of detergent treatment or indicator dye loading. We used this parasite to examine specific effects of ionic dysregulation and advance the understanding of how antimalarials targeting this membrane may work.

## Materials & methods

### Parasite cultivation and transfection

The Dd2 *P. falciparum* clone and its transfected progeny were cultivated in $O^+$ human erythrocytes (University of Virginia Blood Bank) using RPMI 1640 supplemented with 25 mM HEPES, 0.23% NaHCO3, 50 µg/mL hypoxanthine, and 0.5% Microbiological BSA NZ (MP Biomedicals). The modified 4suc:6KCl medium was prepared from individual constituents as described previously [14]; all parasite lines grew normally in this low $Na^+$ medium, allowing direct use for experiments without adaptation.

A transfection plasmid consisting of an 880 bp 5' promoter sequence from the *P. falciparum* 50S ribosomal protein L2 (PF3D7_1132700) followed by a minimal rat *trpv1* gene from the pFastbac1-rTRPV1 vector [15] and a C-terminal 3xHA epitope tag was constructed by InFusion cloning (Takara Bio) into pXL-BacII-hDHFR [16]. The final plasmid, pXL-rTRPV1-HA, was confirmed with restriction digestion and DNA sequencing prior to co-transfection with the pHTH helper plasmid as previously described [17]. These parasites were then selected with 1.5 nM WR99210 (Jacobus Pharmaceuticals) to obtain a transfection pool that was subsequently cloned by limiting dilution to obtain the Dd2-*TRP* line [18]. PCR and Southern blotting were used to confirm the *trpv1* expression cassette and genomic integration.

## Southern blotting

Genomic DNA was extracted using Quick-gDNA™ MiniPrep kit (Zymo Research), quantified on a NanoDrop 2000 spectrometer (ThermoFisher) and digested with PacI. Genomic DNA from indicated parasite clones (5 ug) was resolved along with 2 ng transfection plasmid on a 0.7% agarose gel, acid-depurinated and transferred onto charged Hybond N$^+$ membrane (GE Healthcare) by overnight blotting. A DIG labeled-DNA probe complementary to *hDHFR* was prepared using the hDHFR_Probe-F and hDHFR_Probe-R primers (S1 Table). After crosslinking and prehybridization of the membrane with DIG-Easy Hyb (Roche), the labeled probe was added and hybridized overnight at 42˚C. The blot was washed with low stringency buffer (0.1xSSC/0.1%SDS) at 50˚C and blocked. Probe binding was then detected with anti-digoxigenin-AP Fab fragments (Roche) at a dilution of 1:10,000 and CDP-Star substrate (Roche).

## Membrane fractionation and immunoblot assays

Trophozoite-infected erythrocytes were enriched by the percoll-sorbitol method, washed and lysed in ice-cold hypotonic lysis buffer (7.5mM NaHPO$_4$, 1mM EDTA, pH 7.5) with 1 mM phenylmethylsulfonyl fluoride (PMSF) before ultracentrifugation (100,000 x *g*, 4˚C, 1hr). The supernatant was collected as the soluble fraction; the pellet was further fractionated by resuspension and incubation in 100 mM Na$_2$CO$_3$, pH 11 for 30 min at 4˚C. Ultracentrifugation as above yielded supernatant and pellet fractions representing carbonate-extracted peripheral and integral membrane proteins, respectively. Both fractions were neutralized with 1/10 volume 1 M HCl. All samples were solubilized in Laemmli sample buffer containing 6% SDS.

Proteins were separated by SDS-PAGE (4–15% Mini-Protean TGX gel, Bio-Rad) and transferred to nitrocellulose membrane. After blocking with 5% skim milk in PBS, a primary mouse anti-HA antibody (Sigma-Aldrich) was applied at 1:1000 dilution for overnight incubation at 4˚C. The blot was washed, exposed to HRP-conjugated anti-rabbit secondary antibody (Sigma-Aldrich) with Clarity Western ECL substrate (Bio-Rad) for visualization on Hyblot Xray film. Immunoblots are representative of > 3 trials each.

## Immunofluorescence and immuno-electron microscopy

TRVP1 expression and localization were visualized on thin smears of infected cell cultures after fixation in chilled acetone:methanol at a 1:1 ratio for 5 min. Slides were then blocked with 3% skimmed milk in 1x PBS for 1 h at RT before probing with primary antibodies. The slide was incubated with mouse anti-EXP2 mAb (European Malaria Reagent Repository) in 1:1000 dilution and rabbit anti-HA antibody (AbCam) at 1:100 dilution for overnight at 4˚C. After washing with ice-cold 1x PBS, the slides were incubated with 2 μg/μL 4',6-diamidino-2-phenylindole (DAPI) and secondary antibodies (goat anti-mouse AF488 and goat anti-rabbit AF594) for 30 min at RT. Slides were washed and mounted with Prolong Diamond anti-fade

mountant (Molecular Probes) before visualization under a 64x oil immersion objective on a Leica SP5 confocal microscope and processing with Leica LAS X software.

For transmission EM immunolocalization studies, enriched trophozoite-stage infected cells were fixed with 4% paraformaldehyde/0.05% glutaraldehyde (Polysciences Inc.) in 1x PBS for 1 h at 4˚C. Samples were then embedded in 10% gelatin and infiltrated overnight with 2.3 M sucrose/20% polyvinyl pyrrolidone in 100 mM PIPES, 0.5 mM MgCl$_2$, pH 7.2 at 4˚C. Samples were trimmed, frozen in liquid nitrogen, and sectioned with a Leica Ultracut UCT7 cryo-ultra-microtome (Leica Microsystems Inc.). Ultrathin sections of 50 nm were blocked with 5% fetal bovine serum/5% normal goat serum for 30 min and subsequently incubated with rabbit anti-HA antibody (Sigma, St Louis, MO) for 1 h at room temperature. Following washes in block buffer, sections were incubated with secondary anti-rabbit IgG antibody conjugated to 18 nm colloidal gold (Jackson ImmunoResearch Laboratories, Inc.) for 1 h. Sections were stained with 0.3% uranyl acetate/2% methyl cellulose and viewed on a JEOL 1200 EX transmission electron microscope (JEOL USA Inc.) equipped with an AMT 8-megapixel digital camera and AMT Image Capture Engine V602 software (Advanced Microscopy Techniques, Woburn, MA). All labeling experiments were conducted in parallel with controls omitting the primary antibody.

## Growth inhibition assays

*In vitro* parasite growth inhibition studies with inhibitors were performed in 96-well format as described previously [19]. Sorbitol-synchronized ring stage parasites were adjusted to 1% parasitemia by addition of uninfected erythrocytes, washed twice with RPMI or 4suc:6KCl, and seeded into 96-well plates at 2.5% hematocrit with TRPV1 or PfATP4 inhibitors as indicated. Cultures were maintained at 37˚C for 72 h prior to addition of lysis buffer (20 mM Tris, 10 mM EDTA, 0.016% saponin and 1.6% Triton X-100, pH 7.5) and SYBR Green I nucleic acid gel stain at a 5000x dilution (Invitrogen). Plates were incubated in the dark for at least 30 min before quantification of parasite nucleic acid production with fluorescence measurements (excitation/emission wavelengths, 485/ 528 nm).

Pulse-chase experiments were performed using similar methods to evaluate the effects of short-term TRPV1-induced cation leak at the parasite plasma membrane. Here, synchronous ring stage cultures were exposed to 10 μM capsaicin for 4 or 12 h with matched DMSO controls. At the end of these pulse treatments that were timed to end at 30 h after sorbitol synchronization, all cultures were washed extensively to remove capsaicin before resuming cultivation for a total 72 h incubation. Samples were lysed and nucleic acid production was measured as above.

Growth was normalized to in-plate DMSO and 20 μM chloroquine controls, which corresponded to 100 and 0% growth, respectively. Growth inhibitory IC$_{50}$ values were interpolated from triplicate measurements at each inhibitor concentration.

## Aldolase release assay

The effects of parasite plasma membrane cation leak on membrane integrity were assessed with aldolase release assays as described previously [12]. Sorbitol-synchronized cultures were cultivated to the trophozoite stage, adjusted to 5% parasitemia, and treated with 10 μM capsaicin, 200 nM PA21A092, or DMSO in standard RPMI 1640-based culture medium for 2 h, washed and resuspended in culture medium with indicated saponin concentrations at a 10% hematocrit. After rapid inversions to mix the cell suspension, the freed parasites were harvested by centrifugation, washed and processed for immunoblotting as described above.

Staining for aldolase was used to assess parasite plasma membrane susceptibly to saponin while Exp2 was used as loading control.

### *In vitro* selection of Dd2-*CapR*

The Dd2-*TRP* clone was cultivated with 10 μM capsaicin, producing microscopic sterilization of the culture. With continued cultivation, parasite outgrowth was detected after 15 days. After further growth, limiting dilution cloning was used to generate the Dd2-*CapR* line.

### Structure modeling

Structure modeling was performed using UCSF ChimeraX 1.4 (https://www.rbvi.ucsf.edu/chimerax) [20] and the cryo-EM structure of the minimal rat TRPV1 ion channel (Protein Data Bank accession number 3J5P) [15]. The G683V mutation was modeled using the Dunbrack rotamer library [21] and conservative selection of side chain position based on rotamer prevalence and avoidance of steric clashes. The pore radius was estimated using the HOLE program [22].

### Statistical analysis

Numerical data were analyzed using SigmaPlot 14.5 (Systat). Statistical significance was determined with one-way analysis of variance (ANOVA) with *post hoc* Tukey's multiple comparisons test. Significance was accepted at a threshold $P < 0.05$.

## Results

### TPV1 expression as an integral membrane protein at the *P. falciparum* plasma membrane

We considered several issues and chose the mammalian TRPV1 ligand-gated cation channel for expression at the *P. falciparum* plasma membrane because of its well-characterized transport properties, the availability of multiple ligands for channel activation, negligible ion flux in the absence of activating ligands, known atomic resolution structure, documented expression in other heterologous systems, and permeability to both $Na^+$ and $Ca^{++}$ [23], as proposed substrates for PfATP4 [9, 10]. We chose a minimal rat *trpv1* gene with non-essential regions removed as this yields a stabilized channel structure with demonstrated ligand-responsiveness and transport as well as a reduced gene size that may facilitate expression in the parasite [15, 23]. Importantly, we avoided further modifications, such as those that might facilitate protein trafficking in malaria parasites, to ensure that the channel's functional properties would remain faithful to those established in other organisms.

We also wanted stable integration of the channel gene into the parasite genome for quantitative biochemical studies and used the *piggyBac* transposase system (Fig 1A) [24]. After transfection and PCR to confirm the presence of the *trpv1* cassette (S1 Fig), we performed limiting dilution cloning to obtain the Dd2-*TRP* line. As the *piggyBac* strategy integrates the ITR-flanked region randomly at one or more TTAA recognition sequences in the parasite genome [24], we used Southern blotting to evaluate integration, copy number, and retention of the pXL-rTRPV1-HA plasmid. While the *hdhfr* probe did not hybridize with DNA from the untransfected parent, a single band was recognized from the Pac1-digested DNA of Dd2-*TRP* (S1B Fig); as its size differs from the fragment released by digestion of the transfection plasmid, we confirmed a single integration site without retention of the episome in the Dd2-*TRP* clone.

Immunoblotting using an antibody against the C-terminal HA epitope tag then confirmed expression, detecting a single protein of expected size from Dd2-*TRP* lysates but not from the

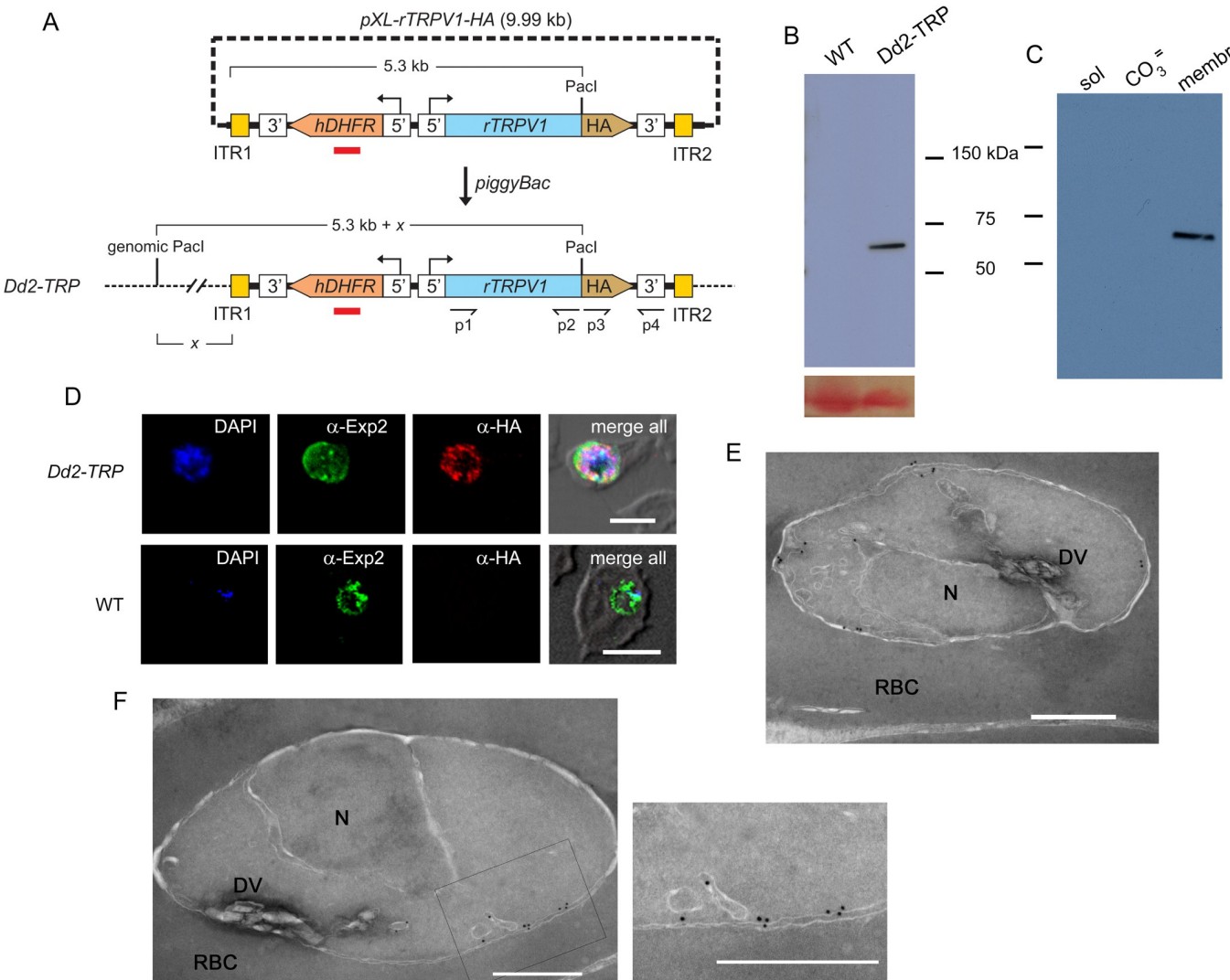

**Fig 1. Expression and localization of the rat TRPV1 channel protein in *P. falciparum*.** (A) Schematic showing *piggyBac* transfection strategy for stable expression of the rat *trpv1* gene product (*rTRPV1*) with a C-terminal HA epitope tag upon integration in the Dd2-*TRP* parasite clone. This strategy integrates the sequence between two inverted terminal repeats (ITR1 and ITR2) into the genome at one or more sites; this region includes the *hDHFR* selection cassette. Primers to confirm retention of the channel gene and the site recognized by the Southern blotting probe (red bar) are shown. (B) Immunoblot of total cell lysates from wild-type (WT) and Dd2-*TRP* lines, probed with mouse anti-HA tag antibody. Ponceau S staining of hemoglobin on the membrane is shown at the bottom as a loading control. (C) Anti-HA immunoblot of match-loaded samples from Dd2-*TRP* after hypotonic lysis and fractionation into soluble, carbonate extractable, and integral membrane pools (sol, $CO_3^=$, and membr, respectively). The expressed TRPV1 protein is integral to parasite membranes. (D) Immunofluorescence antibody (IFA) images of trophozoite-infected cells from indicated parasites, showing labeling with mouse anti-EXP2 and rabbit anti-HA antibodies. TRPV1 localizes within the intracellular parasite and is not exported into the host cell. Scale bar, 5 µm. (E, F) Immunoelectron microscopy images using rabbit anti-HA and a secondary goat anti-rabbit antibody conjugated with 18 nm colloidal gold particles, suggesting TRPV1 trafficking via the parasite ER to the parasite plasma membrane. N, parasite nucleus; DV, digestive vacuole; RBC, red blood cell cytosol; box in panel F, area of zoom in image to right. Scale bars, 500 nm.

parental wild-type line (Fig 1B). Hypotonic lysis and treatment of membranes with $Na_2CO_3$ revealed that the recognized protein is not soluble or released by alkaline treatment, establishing that TRPV1 is expressed exclusively as an integral membrane protein in the parasite (Fig 1C). Indirect immunofluorescence microscopy using anti-HA antibodies recognized the channel protein, excluded export into the host erythrocyte, and suggested localization to the intracellular parasite's plasma membrane (Fig 1D). Immunoelectron microscopy further supported

this localization and also revealed protein in transit via vesicular structures in parasite cytosol (Fig 1E and 1F).

## Rapid parasite death upon ligand-activated cation flux at the parasite plasma membrane

We next examined the effects of focused parasite plasma membrane permeability changes with parasite growth inhibition studies using capsaicin, a TRPV1 ligand unrelated to known PfATP4 inhibitors (Fig 2A and 2B). In 72 h *in vitro* growth assays, capsaicin produced potent and selective killing of Dd2-*TRP* parasites and negligible effect on growth of the wild-type parent (Fig 2B, $P = < 10^{-4}$, Student's *t* test for comparison at 10 μM). Dd2-*TRP* killing was dose-dependent with an $IC_{50}$ of 0.39 ± 0.06 μM, in agreement with the $EC_{50}$ of 0.4 μM reported for rTRPV1 activation on Sf9 cells [25]. Arvanil, a synthetic derivative that more potently activates TRPV1 channels [26], also selectively killed Dd2-*TRP* and worked at >10-fold lower concentrations than capsaicin (Fig 2C, $IC_{50}$ of 31 ± 7 nM), establishing that cation leak through the heterologous ligand-gated channel mediates the observed toxicity. We also used pulse-chase experiments with capsaicin and found that a 12 h pulse produced ~50% killing of Dd2-*TRP* and that a short 4 h application measurably compromised its growth (Fig 2D). This finding parallels the rapid parasite killing by PfATP4 inhibitors under both *in vitro* and *in vivo* conditions [6, 27]. Rapid killing upon conditional permeabilization of the parasite plasma membrane suggests that cation leak has a direct toxic effect on parasite viability.

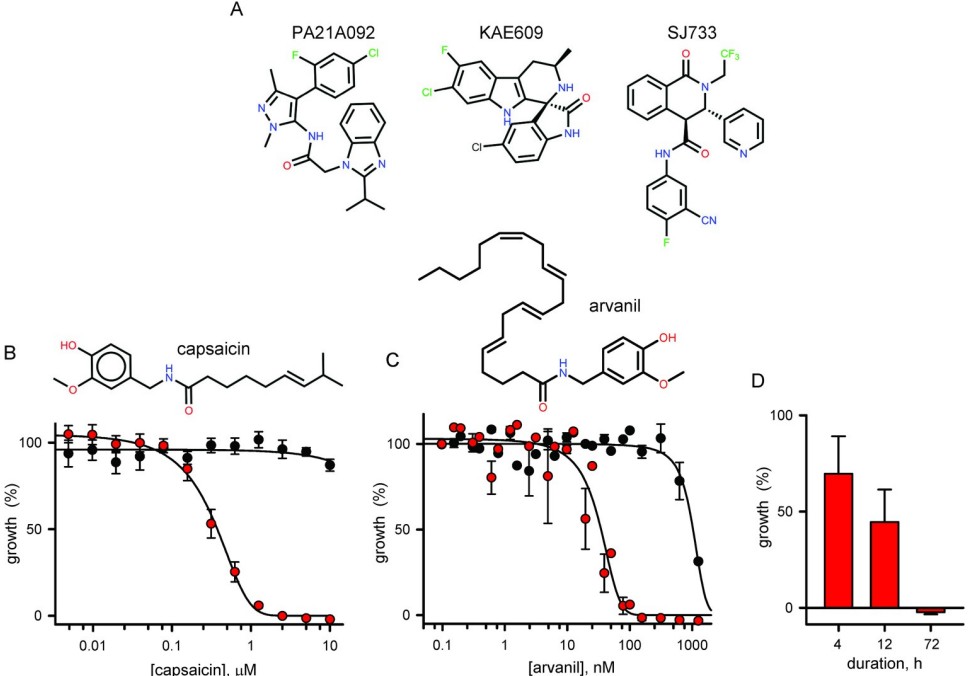

**Fig 2. Conditional permeation of the parasite plasma membrane is lethal.** (A) Structures of indicated antimalarial lead compounds targeting PfATP4. (B) Structure of capsaicin. Bottom panel shows mean ± S.E.M. parasite growth at indicated [capsaicin] for Dd2-*TRP* and wild-type Dd2 (red and black symbols, respectively), determined by combining results from 6–7 independent trials each; Dd2-*TRP* is killed by capsaicin. (C) Structure of arvanil. Plot shows mean ± S. E.M. parasite growth at indicated [arvanil] for Dd2-*TRP* and wild-type Dd2 (red and black symbols, respectively), determined from 3 independent trials each. (D) Mean ± S.E.M. parasite growth over 72 h with a pulse-chase application of 10 μM capsaicin for indicated durations, determined from 3–5 independent trials with normalization to 100% growth for DMSO control and 0% for matched cultures treated with 20 μM chloroquine for 72 h. Reduced growth relative to DMSO control indicates rapid killing with short applications of capsaicin.

## TRPV1 activation increases PPM saponin sensitivity

PfATP4 block and the resulting loss of $Na^+$ and $H^+$ gradients have been linked to compromised integrity of the parasite plasma membrane, as identified through an increased susceptibility to saponin, a cholesterol-dependent detergent [12]. Subsequent studies suggest that $Na^+$ accumulation and/or alkalization in the parasite cytosol leads to inhibition of PfNCR1, a lipid transporter that maintains a low cholesterol content in the PPM [28], but other mechanisms, such as off-target effects of PfATP4 inhibitors, are also possible. We therefore examined whether TRPV1 activation leads to increased saponin susceptibility of the PPM and followed the protocol of Das *et al.* [12], which uses immunoblotting to evaluate leak of soluble aldolase from the parasite compartment upon saponin exposure.

Although initial studies with several batches of saponin failed to reproduce aldolase leak after PfATP4 block in our hands, we successfully reproduced their results with two separate lots of saponin provided by the Vaidya laboratory (Fig 3). We confirmed that a brief 2h treatment with PA21A092, a potent PfATP4 inhibitor, increased saponin susceptibility and released aldolase from the parasite pellet in both wild-type and Dd2-*TRP* parasites while a matched DMSO treatment was without effect (control, Fig 3). A 2 h capsaicin pretreatment replicated this phenotype but was specific for Dd2-*TRP* parasites, indicating that saponin susceptibility results directly from increased permeabilization of the PPM. Interestingly, with the lot of saponin used in most of our experiments, we noticed that the capsaicin-induced alterations were detected only at higher concentrations of saponin than those required after PA21A092 treatment (1% saponin for capsaicin vs. 0.08% for PA21A092 in Fig 3). As a matched DMSO pretreatment did not lead to aldolase leakage, we excluded direct damage of the PPM by this higher saponin concentration and, instead, directly implicated a role of cation uptake in the presumed PPM membrane fragility.

## Efficacy in low $Na^+$ media

Vaidya *et al.* also reported that PfATP4 inhibitors are less effective in parasite growth inhibition studies when parasites are cultivated in media with markedly reduced $Na^+$ levels [7]. When compared to standard RPMI 1640-based medium containing 143 mM [$Na^+$], *P. falciparum* cultivation proceeds at identical rates in a modified medium termed 4suc:6KCl, where

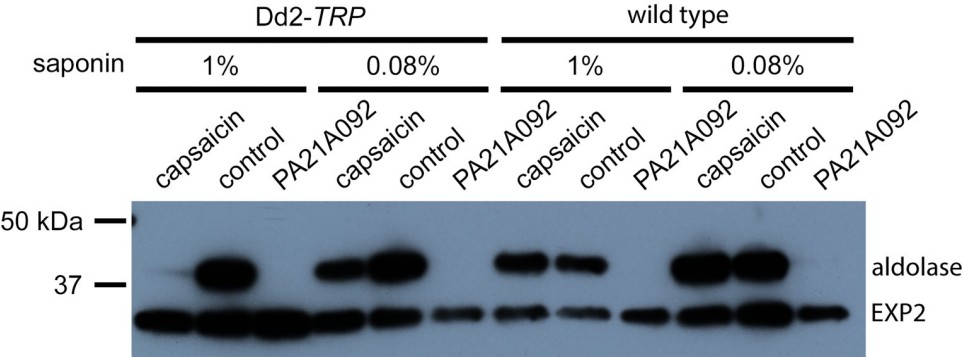

**Fig 3. TRPV1 activation renders the parasite plasma membrane sensitive to saponin, phenocopying PfATP4 block.** Immunoblot showing aldolase retention in trophozoite-stage freed parasites after pretreatment with 10 μM capsaicin, 200 nM PA21A092, or DMSO (control) and exposure to indicated saponin concentrations. Capsaicin pretreatment followed by 1% saponin exposure leads to aldolase release from Dd2-*TRP*, but not the Dd2 wild type parent. Notice that PA21A092-mediated PfATP4 block renders parasite membranes sensitive to lower concentrations of saponin than seen after capsaicin treatment.

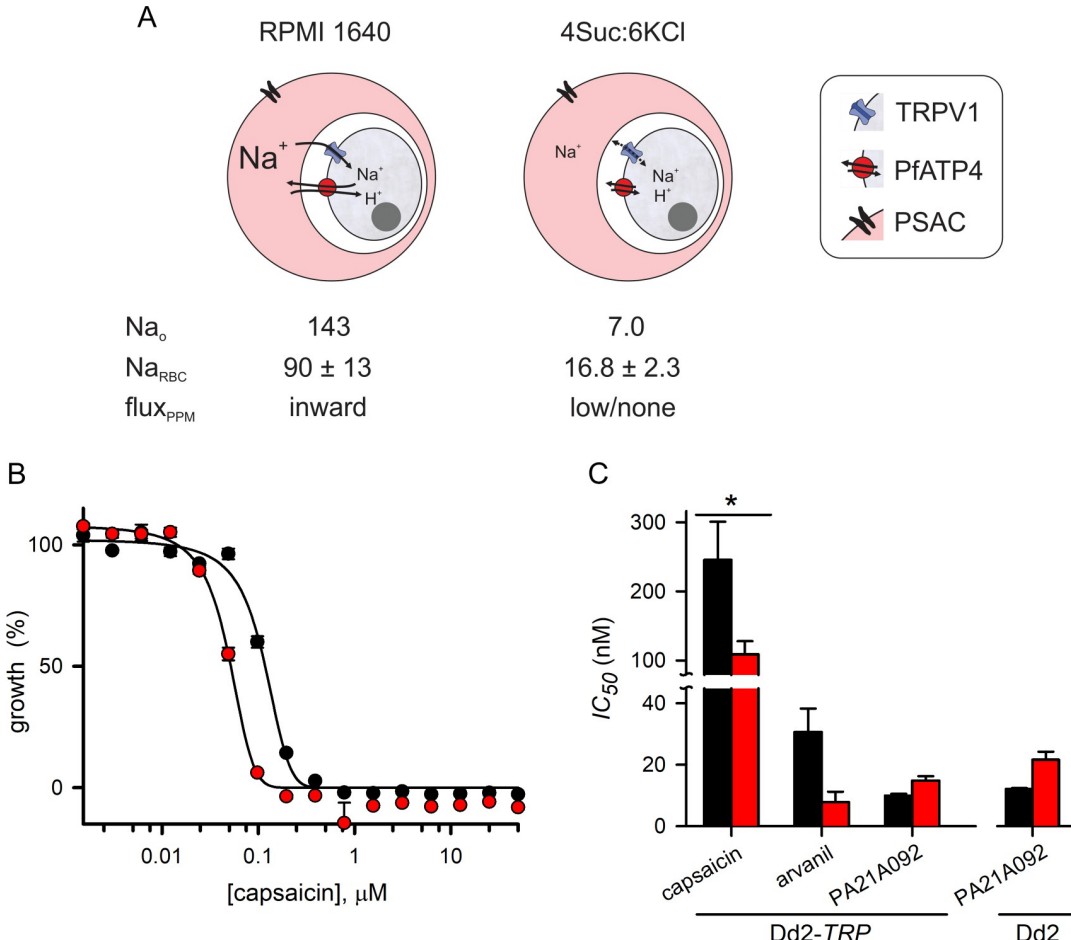

**Fig 4. Growth inhibition studies using low Na$^+$ media.** (A) Schematic showing ion flux when infected erythrocytes are cultivated in media based on the standard RPMI 1640 and modified 4Suc:6KCl formulations. Erythrocyte [Na$^+$] increases in RPMI 1640 medium due to PSAC-mediated influx at the host membrane but remains low in 4Suc:6KCl, yielding a negligible gradient for uptake at the parasite plasma membrane. Arrow lengths on TRPV1 indicate Na$^+$ gradient; those on PfATP4 indicate expected requirement for pump-mediated extrusion. Values at bottom tabulated from [14] with similar increases in erythrocyte Na$^+$ reported previously [29]. (B) Capsaicin dose response for Dd2-*TRP* killing in RPMI 1640 medium and 4Suc:6KCl (black and red symbols, mean ± S.E.M. of triplicate measurements from a matched growth experiment). Note the unexpected improvement in killing when Na$^+$ is reduced. (C) Mean ± S.E.M. $IC_{50}$ values from growth inhibition experiments using indicated TRPV1 ligands and PA21A092 in RPMI 1640 and 4Suc:6KCl media (black and red bars, respectively). Only capsaicin produced a significant difference in $IC_{50}$ values for the two media. *, $P = 0.05$ (one-way ANOVA with Tukey's multiple comparisons); $n$ = 3–7 independent matched trials each.

[Na$^+$] is reduced to 7 mM through replacement with an optimal combination of K$^+$ and sucrose [14]. In this medium, the erythrocyte cytosolic [Na$^+$] remains unchanged from the ~ 10 mM levels in uninfected erythrocytes, contrasting with marked increases that occur in physiological media or human plasma (Fig 4A) [14, 29]. Physiological increases in erythrocyte Na$^+$ content during parasite maturation result from slow influx via the plasmodial surface anion channel (PSAC) [14, 30]; unabated growth in 4suc:6KCl indicate that the increased erythrocyte Na$^+$ is not required by the intracellular parasite. Importantly, cultivation in 4suc:6KCl reduces the Na$^+$ gradient at the PPM, rendering PfATP4-mediated Na$^+$ extrusion less critical for intracellular parasite survival (Fig 4A). The reduced efficacy of PfATP4 inhibitors in this low Na$^+$ medium would therefore be consistent with the proposed essential role of PfATP4 in Na$^+$ extrusion to maintain Na$^+$ homeostasis in parasite cytosol. In their studies

with several PfATP4 inhibitors, increases in the half-maximal concentrations for parasite growth inhibition, $IC_{50}$ values, increased by a modest 3–4 fold [7] and statistical significance was not reported.

We therefore performed growth inhibition studies with TRPV1 ligands and PA21A092. Interestingly, capsaicin and arvanil exhibited increased efficacy in 4suc:6KCl medium (Fig 4B and 4C), rather than the anticipated reduced efficacy. As observed by Vaidya *et al.*, we noted a trend toward an increased $IC_{50}$ for the PfATP4 inhibitor PA21A092 in low $Na^+$, but this was not statistically significant in experiments with either Dd2-*TRP* or its wildtype parent (Fig 4C).

### Rapid selection of a resistant mutant with an altered TRPV1 channel pore

Under *in vitro* selection, *P. falciparum* cultures quickly acquire resistance to pyrazoleamides, spiroindolones, and (+)-SJ733, advanced antimalarial lead compounds with distinct scaffolds (Fig 2A). Each of these selected mutants carries distinct mutations in PfATP4, implicating action on this target. Biochemical studies with these mutants suggest that the pump retains its physiological transport activity despite inhibitor addition, possibly through mutations that compromise drug binding.

We therefore used *in vitro* selection of Dd2-*TRP* cultures with capsaicin and detected outgrowth of resistant mutants in 15 days, paralleling rapid selection of resistant mutants with the above PfATP4 inhibitors. Limiting dilution cloning yielded Dd2-*CapR*, a line that had fully lost its sensitivity not only to capsaicin, but also to arvanil (Fig 5A and 5B). This resistant mutant had an unchanged sensitivity to PA21A092 (Fig 5C), consistent with action on independent transporters at the PPM. This finding also suggests that selected mutant does not have a greater tolerance for cation leak into parasite cytosol.

Southern blotting and immunoblotting experiments revealed that Dd2-*CapR* did not lose the *trpv1* gene and that it continues to express the channel protein at comparable levels (S1B Fig and Fig 5D). DNA sequencing of the *trvp1* gene revealed a single nucleotide polymorphism, yielding a G683V mutation in the protein. Examination of the rat TRPV1 structure reveals that this mutation lines the pore at its cytoplasmic end (Fig 5F), near the lower gate [15]. Binding of capsaicin and other vanilloid agonists is thought to displace a phosphoinositide lipid that interacts with the TRPV1 channel within the membrane, leading to pore opening through a rearrangement that opens the lower gate. We therefore modeled the G683V mutation and found that it constricts the pore just distal to this lower gate (Fig 5G). Notably, G683 is distant from the site of vanilloid agonist binding, so this mutation is not likely to alter binding of capsaicin, arvanil and other agonists. This contrasts with the presumed mechanism of escape from block by PfATP4 inhibitors, where mutations are thought to reduce inhibitor binding and action [5–7].

## Discussion

While all eukaryotic cells must regulate their intracellular ion concentrations, bloodstream malaria parasites face additional constraints because they must achieve this regulation while growing in erythrocyte cytosol. Importantly for *P. falciparum*, the host compartment changes dramatically from a low $Na^+$/high $K^+$ environment to one that is high $Na^+$/low $K^+$ due to host ion remodeling by PSAC-mediated leaks (Fig 4A) [29, 30]. Transport activities at the parasite plasma membrane (PPM) are essential for adapting to these changes, as highlighted by the discovery of potent PfATP4 inhibitors that interfere with parasite ion homeostasis [5–7]. Here, we used a novel approach, expression of a heterologous ligand-gated ion channel at the PPM, to examine the importance of ion regulation at this membrane and to establish a direct link

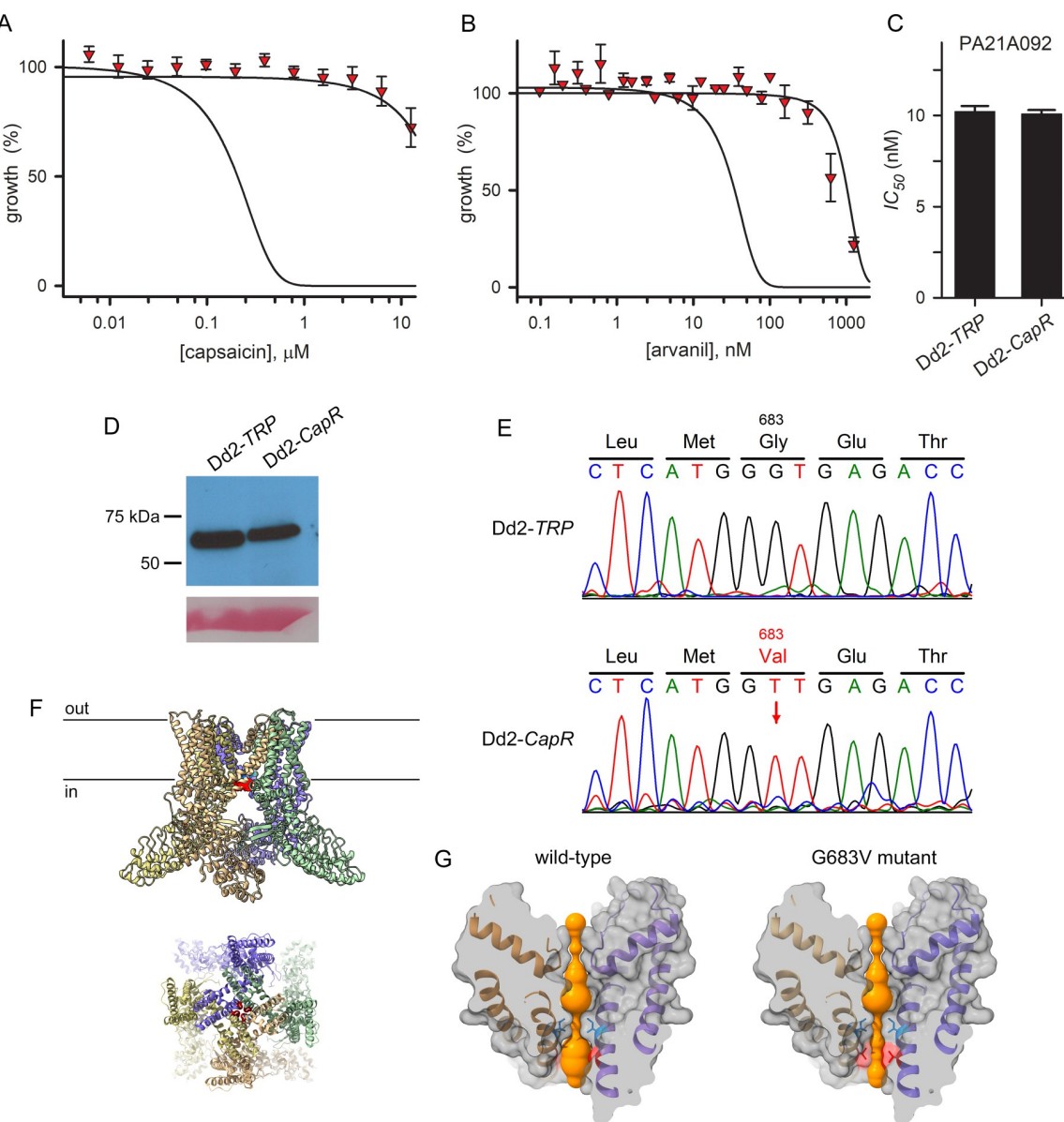

**Fig 5. Rapid selection of a mutant that avoids cation leak at the PPM.** (A, B) Mean ± S.E.M. growth of Dd2-*CapR* at indicated [capsaicin] and [arvanil] (red triangles), determined by combining results from 3 independent trials each. In each panel, black lines represent best fits for dose-response data in matched experiments using wild-type Dd2 and Dd2-*TRP* (top and bottom) to single sigmoidal decay. Note that Dd2-*CapR* sensitivity to both ligands is indistinguishable from that of the wild-type parent, indicating a quantitatively complete loss of efficacy against the engineered transfectant. (C) Mean ± S.E.M. half-maximal growth inhibition concentrations ($IC_{50}$) for indicated parasite lines by the PfATP4 inhibitor PA21A092. Selection did not alter the parasite's susceptibility to PfATP4 block. (D) Anti-HA immunoblot showing TRPV1 expression in Dd2-*CapR*. Bottom, Ponceau S loading control. (E) Sequence chromatograms from the Dd2-*TRP* transfectant and the selected Dd2-*CapR* clone, showing the single nucleotide polymorphism that alters G683 (rat TRPV1 numbering) to valine. (F) Modeled cartoon structure of rat TRPV1 with the G683V mutation, based on the cryo-EM structure [15]. This mutation (red) is at the lower TRPV1 gate and lines the ion conduction pore; side and top views of the membrane-embedded channel are shown (top and bottom images). (G) Cutaway side views of wild-type and mutant channel pores. Blue sticks represent the sidechain of I679, which forms the lower gate in the wild-type channel [43]; the position of G683 and the larger valine sidechain in the mutant are shown in red cartoons, sidechain sticks, and associated surface patches. The solvent-accessible pathway was mapped using the HOLE program and its surface is shown in orange. Note the narrowed pore in the mutant (bottom), suggesting that cation flux is prevented by an occluded pore.

between uptake and downstream sequela. Use of a ligand-gated channel permitted conditional activation of ion leak at the PPM and revealed that rapid parasite killing results directly from unregulated cation influx. We also determined that cation influx alone is sufficient to produce rapid changes in PPM lipid composition that render it susceptible to saponin, a cholesterol-dependent detergent that normally does not permeabilize the PPM. *In vitro* selection with capsaicin, a canonical TRPV1 agonist, produced a resistant mutant with a defective pore, providing conclusive evidence for essential cation regulation in the intracellular parasite.

Although PfATP4 inhibitors also render the PPM susceptible to saponin through a process linked to a recently identified lipid-transporter at this membrane [28], prior studies were unable to directly attribute the membrane changes to PPM cation flux because of possible off-target effects of these antimalarial agents. We addressed this concern and implicated a direct effect of ion flux because capsaicin does not produce saponin-susceptibility in wild-type parasites, but instead requires TRPV1 expression and activation of a conditional cation leak. Prior studies suggest that PfATP4 inhibition simultaneously increases [Na$^+$] and pH in the parasite cytosol, raising the additional question of whether one or both of these changes mediates altered PfNCR1 function and saponin susceptibility. Maduramicin, an anticoccidial drug that functions as an ionophore, has been used to implicate Na$^+$ based on presumed selectivity for Na$^+$ over H$^+$ [12, 31]. Because maduramicin forms complexes with many monovalent and divalent cations [32], its cation rank order selectivity for cation transport across parasite membranes remains unclear. Moreover, because this ionophore may also partition into the membranes of various parasite organelles, saponin susceptibility and parasite killing by this agent should be interpreted cautiously.

We note that TRPV1 is also non-selective, transporting Ca$^{++}$, Na$^+$, and H$^+$ flux in electrophysiological studies [13, 33]. Although TRPV1 has a high Ca$^{++}$ permeability, parasite killing by TRPV1 ligands is not likely to result from Ca$^{++}$ influx at the PPM because Ca$^{++}$ is maintained at submicromolar free concentrations in both the host and parasite compartments by a high affinity Ca$^{++}$ ATPase pump at the cell surface [34–36]. In contrast, host cytosol Na$^+$ increases dramatically as the parasite matures, creating a large inward electrochemical gradient at the PPM. Thus, TRPV1 activation in Dd2-*TRP* will raise parasite [Na$^+$]. Robust metabolic acid production in the parasite combined with rapid equilibration of acid equivalents between erythrocyte cytosol and the extracellular medium suggests that H$^+$ has an outward gradient at the PPM [37, 38], but this has not been experimentally established. If correct, TRPV1 activation will also raise pH within parasite cytosol, as seen with PfATP4 inhibition [10]. Our study is therefore unable to determine whether increased parasite Na$^+$, reduced H$^+$, or both are required to trigger the observed changes in PPM lipid composition. More fundamentally, the precise mechanisms through which altered ion contents lead to dysregulated lipid content as well as other changes within infected cells that culminate in rapid parasite demise remain mysterious and are central to advancement of PfATP4 inhibitors as antimalarial drug leads.

While examining saponin susceptibility of the PPM, we observed quantitative differences in the effects of TRPV1 activation and PfATP4 inhibition. Notably, while both capsaicin and PA21A092 compromised aldolase retention in saponin-released parasites, a higher saponin concentration was required with capsaicin (1% saponin vs 0.08% for PA21A092 in Fig 3), despite application at levels that fully active TRPV1. We considered several explanations for this difference. First, because the untransfected parental line is not rendered saponin sensitive by capsaicin (Fig 3), this ligand does not inhibit PfATP4-mediated Na$^+$ efflux. Sustained PfATP4 activity upon TRPV1 activation by capsaicin may partially mitigate Na$^+$ influx and account for the lower saponin sensitivity. Second, an intriguing possibility is that TRPV1 activation and PfATP4 inhibition produce different changes in parasite cation concentrations. Notably, because TRPV1 is less selective, its activation is expected to dissipate not only the Na$^+$

and $H^+$ gradients at the PPM, but also any $Ca^{++}$ gradients at this membrane. Although $Ca^{++}$ concentrations are low in both host and parasite compartments, this ion's key roles in intracellular parasite development suggest it too must be regulated at the PPM [39, 40]. Loss of possible PPM $Ca^{++}$ gradients upon TRPV1 activation may therefore account for the differing effects of capsaicin and PA21A092 in our aldolase release experiments. Finally, there are several less interesting explanations for the differing saponin sensitivities that should be kept in mind—the unknown relative abundance of TRPV1 and PfATP4 at the PPM, the absolute cation flux rates through these transporters, and possible effects of pyrazoleamides on the PPM other than those resulting from increases in parasite $Na^+$.

Batch-to-batch variation in saponin's effectiveness, as we observed, further complicate examination of differing effects of TRPV1 activation and PfATP4 inhibition. This batch-dependent effect presumably results from variable content of sapogenins or other surfactants in this crude plant extract. Nevertheless, the saponin susceptibility's link to PfATP4 block and our demonstration that it results directly from cation leak render this assay useful for uncovering the precise mechanisms that underly the actions of advanced antimalarial compounds.

We selected random integration of the *trpv1* expression cassette via the *piggyBac* system in our transfections because this approach can sometimes yield clonal lines with more than one integration event [24]. Varying copy numbers would have allowed TRPV1 dose-effect studies and provided insights into the above quantitative differences in saponin susceptibilities; unfortunately, we obtained only clones with single integration events (S1B Fig). Future studies using this and other heterologous ligand-gated channels to examine ion regulation in the intracellular parasite should consider this possible advantage and weigh it against the advantages of site-directed integration via CRIPSR-Cas9 gene insertion.

An important limitation of prior biochemical studies of PPM transport is that they have required detergent release of the intracellular parasite, followed by loading of $Na^+$ and $H^+$ indicator dyes with AM esters [6, 10, 12]. Because these dyes are poorly soluble, these studies used Pluronic F-127, a second detergent, to facilitate dye loading into the freed parasite. Moreover, AM ester hydrolysis, as required for trapping and activation of the indicator within parasite cytosol, releases formaldehyde and is known to be cytotoxic [41, 42]. Although PfATP4-mediated $Na^+/H^+$ exchange has been confirmed with other methods including measurement of ion-dependent ATPase activity in membrane preparations [10], these concerns have limited the interpretation of prior cell-based transport studies and prevented complete understanding of how PfATP4 inhibitors act against the intracellular parasite. In contrast to the infected cell host membrane and the parasitophorous vacuolar membrane (PVM), the PPM is also not amenable to patch-clamp, preventing direct study of ion transport at this membrane [8]. Conditional and selective permeabilization of the PPM through expression of a ligand-gated channel, as performed here, circumvents these concerns and provides a novel approach to examine and perturb ion regulatory mechanisms at inaccessible membranes. Biochemical studies of cation flux using Dd2-*TRP* and clones expressing other ligand-gated channels, using indicator dyes and other approaches, will help guide development of improved antimalarials targeting ion homeostasis at the PPM.

## Supporting information

**S1 Table. Primers used in this study.**
(XLSX)

**S1 Fig. Transfection with pXL-rTRPV1-HA.** (A) Ethidium-stained gel showing PCR checks for retention of the *trpv1* gene in the *piggyBac* transfectant. Lanes show PCR products using indicated primers for the transfectant line (Dd2 +TRPV1), but not with the untransfected

parental control (Dd2). Primer positions are indicated in Fig 1A; sequences are provided in S1 Table. Expected sizes: *p1-p2*, 1949 bp; *p3-p4*, 130 bp. (B) Southern blotting showing DNA from indicated parasites or the pXL-rTRPV1-HA transfection plasmid control, each digested with PacI. While the *hDHFR* probe does not hybridize to DNA from the wild-type Dd2 parent, both Dd2-*TRP* and Dd2-*CapR* yield a single band (size ~6 kbp) distinct from the 9.99 kbp seen with the transfection plasmid, indicating a single detected integration into the parasite genome. As the 9.99 kbp band is not detected in Dd2-*TRP*, this clone does not carry residual episomes. An additional smaller band (< 4.9 kbp) in the plasmid control reflects undigested, supercoiled plasmid. An increased exposure image of the same blot is shown on the right to confirm probe specificity and an unchanged band in Dd2-*CapR*.
(TIF)

**S1 Raw images.**
(PDF)

## Acknowledgments

We thank David Julius for providing the pFastbac1-rTRPV1 vector, David Jacobus for WR99210, Ahkil Vaidya and members of his lab for providing saponin and guidance on aldolase release experiments, Phil Cruz for help with structure modeling, and Wandy Beatty for help with immuno-EM experiments.

## Author Contributions

**Conceptualization:** Mariame Sylla, Sanjay A. Desai.

**Formal analysis:** Mariame Sylla, Ankit Gupta, Jinfeng Shao, Sanjay A. Desai.

**Investigation:** Mariame Sylla, Ankit Gupta, Jinfeng Shao.

**Supervision:** Sanjay A. Desai.

**Validation:** Mariame Sylla, Ankit Gupta, Jinfeng Shao, Sanjay A. Desai.

**Writing – original draft:** Mariame Sylla, Sanjay A. Desai.

**Writing – review & editing:** Mariame Sylla, Ankit Gupta, Jinfeng Shao, Sanjay A. Desai.

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
