## [Decision Letter · Decision Letter 0]

1 Feb 2023

PONE-D-22-32957Conditional permeabilization of the P. falciparum plasma membrane in infected cells links cation influx to reduced membrane integrityPLOS ONE

Dear Dr. Desai,

Thank you for submitting your manuscript to PLOS ONE. After careful consideration, we feel that it has merit but does not fully meet PLOS ONE’s publication criteria as it currently stands. Therefore, we invite you to submit a revised version of the manuscript that addresses the points raised during the review process.

We look forward to receiving your revised manuscript.

Kind regards,

Jude M Przyborski

Academic Editor

PLOS ONE

Journal Requirements:

Additional Editor Comments:

Dear Sanjay,

Once again apologies for the long review process.

As you will see, both reviewers are very positive about your manuscript, however reviewer 1 would appreciate a few minor changes.

I think if you can incorporate these into a revised manuscript, then your piece should be suitable for publication.

Best regards,

Jude

Reviewers' comments:

Reviewer's Responses to Questions

**Comments to the Author**

1. Is the manuscript technically sound, and do the data support the conclusions?

Reviewer #1: Yes

Reviewer #2: Yes

2. Has the statistical analysis been performed appropriately and rigorously? 

Reviewer #1: Yes

Reviewer #2: Yes

3. Have the authors made all data underlying the findings in their manuscript fully available?

Reviewer #1: Yes

Reviewer #2: Yes

4. Is the manuscript presented in an intelligible fashion and written in standard English?

Reviewer #1: Yes

Reviewer #2: Yes

5. Review Comments to the Author

Reviewer #1: The authors present an elegant an innovative approach to studying the role of malaria ATP4 and the mechanism of action of ATP4 inhibitors. Plasmodium ATP4 has emerged as a key target of pipeline antimalarials, with several distinct classes. The precise mechanism or parasite killing is unclear but ATP inhibitors are known to cause key changes in cellular pH, Na+ levels and plasma membrane composition. This work uses expression of a ligand gated cation channel as a tool to dissect these changes. The resultant transgenic parasites demonstrate selective sensitivity to TRPV1 agonists which result in parasite killing and changes in saponin sensitivity analogous to that seen for ATP4 inhibitors. The work is well written and I have only minor suggestions, mostly surrounding additional discussion of rationale and findings.

Minor comments

Technical approach - I am not sure why the authors opted for random integration using the piggyBac system – random integration and potential for variable copy number are big disadvantages, which could be avoided using various Cas9 or even AttB integrase type approaches. These disadvantages are well mitigated by the follow up work, including appropriate genotyping and phenotypic controls. The fact that the integration site is unknown will make these specific lines less attractive for others to use. That said, this does not affect the success of this work in validating the fundamental approach and of course similar lines could be readily remade targeting a known locus. I think the parasite lines will be of broader interest and the methodological development itself provides significant impact, so some discussion of these tool parasites, for example for steps to characterise their integration site/generate separate lines and some additional use case examples would be a good addition.

Genotyping - The combined genotypic data of PCRs Southerns, and IFA is convincing. If available the Southern blots should include a positive control probe for the DD2 wild type to confirm even DNA loading.

Localisation - The EM and IFA images are consistent with membrane localisation, but further IFA images from other parasite stages (particularly rings and late trophozoites) would be informative. It might be easier to discern clear parasite membrane localisation in earlier stages.

Saponin source - The fact that clear differences in saponin reactivity were seen between different preparation, could the authors comment on that in the discussion. Is there a difference in source/purity etc which could account for this?

Biochemical characterisation – Although authors discuss the draw backs of some of approaches to measure changes in pH/cation flux in live parasites, it would clearly be an important next step to compare findings with these tool parasites. I don’t think it is necessary to add such experimental data here, but it would be appropriate to lay out next steps for biochemical analysis of the TRPV1 lines in the discussion.

All the best,

Rob Moon

Reviewer #2: The manuscript “Conditional permeabilization of the P. falciparum plasma membrane in infected cells links cation influx to reduced membrane integrity” by Sylla et al. is a pleasure to read. It describes a very original approach to control sodium permeability of the blood stage malaria parasite plasma membrane by introduction of the TRPV1 channel. This will be a great tool for the research community to explore processes related to the ion homeostasis of the parasite. In the present work the authors used the engineered parasite to test a previously proposed model connecting the function of a PfATP4, a sodium and proton exchange pump, directly with lipid homeostasis. The study is very elegant and every alternative hypothesis that I was able to think of was addressed in the discussion. In my opinion the manuscript can be published as is.

6. PLOS authors have the option to publish the peer review history of their article (what does this mean?). If published, this will include your full peer review and any attached files.

Reviewer #1: **Yes: **Robert W. Moon

Reviewer #2: No

---

## [Author Response · Author response to Decision Letter 0]

3 Mar 2023

See attached Response to Reviewers file

---

## [Editor Report · Decision Letter 1]

16 Mar 2023

Conditional permeabilization of the P. falciparum plasma membrane in infected cells links cation influx to reduced membrane integrity

PONE-D-22-32957R1

Dear Dr. Desai,

We’re pleased to inform you that your manuscript has been judged scientifically suitable for publication and will be formally accepted for publication once it meets all outstanding technical requirements.

Kind regards,

Jude M Przyborski

Academic Editor

PLOS ONE
---

## [Editor Report · Acceptance letter]

27 Mar 2023

PONE-D-22-32957R1 

Conditional permeabilization of the *P. falciparum* plasma membrane in infected cells links cation influx to reduced membrane integrity 

Dear Dr. Desai:

I'm pleased to inform you that your manuscript has been deemed suitable for publication in PLOS ONE. Congratulations! Your manuscript is now with our production department. 

Kind regards, 

on behalf of

Dr Jude M Przyborski 

Academic Editor

PLOS ONE